# Distinct and Overlapping Functions of *Miscanthus sinensis* MYB Transcription Factors SCM1 and MYB103 in Lignin Biosynthesis

**DOI:** 10.3390/ijms222212395

**Published:** 2021-11-17

**Authors:** Philippe Golfier, Olga Ermakova, Faride Unda, Emily K. Murphy, Jianbo Xie, Feng He, Wan Zhang, Jan U. Lohmann, Shawn D. Mansfield, Thomas Rausch, Sebastian Wolf

**Affiliations:** 1Centre for Organismal Studies Heidelberg, Heidelberg University, 69120 Heidelberg, Germany; philippe.golfier@alumni.uni-heidelberg.de (P.G.); olga.ermakova@cos.uni-heidelberg.de (O.E.); hefeng@qibebt.ac.cn (F.H.); zhangwan1105@hotmail.com (W.Z.); jan.lohmann@cos.uni-heidelberg.de (J.U.L.); 2Department of Wood Science, University of British Columbia, Vancouver, BC V6T 1Z4, Canada; farideu@gmail.com (F.U.); emily.ka.murphy@gmail.com (E.K.M.); shawn.mansfield@ubc.ca (S.D.M.); 3Key Laboratory of Genetics and Breeding in Forest Trees and Ornamental Plants, College of Biological Sciences and Technology, Beijing Forestry University, No. 35, Qinghua East Road, Beijing 100083, China; jbxie@bjfu.edu.cn; 4Center for Plant Molecular Biology (ZMBP), University of Tübingen, Auf der Morgenstelle 32, 72076 Tübingen, Germany

**Keywords:** cell wall, lignin, biomass, transcriptional regulation, Miscanthus, MYB

## Abstract

Cell wall recalcitrance is a major constraint for the exploitation of lignocellulosic biomass as a renewable resource for energy and bio-based products. Transcriptional regulators of the lignin biosynthetic pathway represent promising targets for tailoring lignin content and composition in plant secondary cell walls. However, knowledge about the transcriptional regulation of lignin biosynthesis in lignocellulosic feedstocks, such as Miscanthus, is limited. In Miscanthus leaves, *MsSCM1* and *MsMYB103* are expressed at growth stages associated with lignification. The ectopic expression of *MsSCM1* and *MsMYB103* in *N. benthamiana* leaves was sufficient to trigger secondary cell wall deposition with distinct sugar and lignin compositions. Moreover, RNA-seq analysis revealed that the transcriptional responses to *MsSCM1* and *MsMYB103* overexpression showed an extensive overlap with the response to the NAC master transcription factor MsSND1, but were distinct from each other, underscoring the inherent complexity of secondary cell wall formation. Furthermore, conserved and previously described promoter elements as well as novel and specific motifs could be identified from the target genes of the three transcription factors. Together, MsSCM1 and MsMYB103 represent interesting targets for manipulations of lignin content and composition in Miscanthus towards a tailored biomass.

## 1. Introduction

Lignocellulosic biomass, which largely exists in the form of plant secondary cell walls (SCW), holds enormous potential as a renewable feedstock for a sustainable economy [1]. Despite various biorefining applications, the processing of lignocellulosic biomass into bio-based products and energy is still hampered by the inherent resistance of cell walls to deconstruction, which is largely conferred by the aromatic polyphenol lignin [1,2]. Lignin associates with the cellulose and hemicellulose network of SCWs, providing mechanical support, rigidity, and hydrophobicity. The mechanical importance of lignin is highlighted by lignin-deficient mutants that manifest dwarfism, collapsed xylem vessels, or higher susceptibility to pathogens, although some of those phenotypes can also be the result of signaling-mediated secondary responses [3]. Due to cell wall recalcitrance, lignin engineering has become a focal point of efforts aiming to alter lignin content, composition, and structure, exploiting its inherent plasticity [4]. Thus, it is important to understand the mechanisms of lignin biosynthesis and its regulation to harness the potential of lignocellulosic biomass as a renewable resource for biorefineries.

Plants of the genus Miscanthus are regarded as some of the most promising energy crops for the production of lignocellulosic biomass in temperate climates. Miscanthus species are rhizomatous, perennial grasses that combine efficient C4 metabolism and favorable leaf morphology, resulting in high yields with modest water and nutrient requirements. Breeding programs have generated novel Miscanthus hybrids to overcome major limitations for adoption into diverse agricultural systems displaying different soil qualities and wide climatic ranges [5]. Moreover, Miscanthus breeding would benefit from molecular markers to facilitate the selection of varieties with an improved biomass yield and/or composition [6]. Genetic resources for Miscanthus, such as an extensive transcriptome database [7] and a recently completed *Miscanthus sinensis* draft genome, *Miscanthus sinensis* v7.1 DOE-JGI, accessible at Phytozome (http://phytozome.jgi.doe.gov/, accessed on 12 November 2021), are now available to explore biological processes with molecular approaches. A detailed, mechanistic understanding of SCW formation has the potential to complement the marker-assisted breeding of novel Miscanthus traits with tailored biomass.

The irreversibility of SCW formation requires a tight regulation of the biosynthesis of cellulose, hemicellulose, and lignin. A sophisticated multi-tiered network of NAM, ATAF1,2, CUC (NAC), and MYB transcription factors (TFs) coordinately integrate developmental and environmental aspects [8,9,10,11]. In Arabidopsis, the closely related NAC SECONDARY CELL WALL THICKENING PROMOTING FACTOR (NST)1 and 2, SECONDARY WALL-ASSOCIATED NAC DOAMAIN1 (SND1)/NST3, and VASCULAR-RELATED NAC DOMAIN (VND) 6 and 7 have been identified as master switches regulating SCW formation, including the regulation of biosynthetic pathways for cellulose, hemicellulose, and lignin [11,12,13,14,15], as well as controlling the patterned deposition of SCW components [15]. A cis regulatory element, SNBE ((T/A)NN(C/T) (T/C/G)TNNNNNNNA(A/C)GN(A/C/T)(A/T)), has been identified as a binding site for secondary cell wall NACs in the promoter region of direct targets (Zhong et al., 2010). Downstream of NAC TFs, MYB46 and MYB83 are situated as second-tier regulators and were shown to act redundantly during SCW formation [16,17]. MYB46 was shown to bind to the SMRE element (ACC(A/T)A(A/C)(T/C)) in target promoters (Zhong et al., 2011). A large number of lower-tier members such as MYB20, MYB42, MYB43, MYB58, MYB63, MYB83, MYB85, and MYB103 are often targets of both the second-tier MYB46 and 83 as well as of the NAC master switches, and are assumed to transcriptionally finetune SCW formation, particularly lignin biosynthesis [18,19,20]. However, lower-tier MYB factors from grasses seem to regulate additional secondary cell wall components along with lignin [21,22]. For example, MYB103 in Arabidopsis has been described to specifically affect lignin composition without affecting the total lignin and cellulose content [23], whereas MYB103 orthologues from grass species seem to regulate a much broader spectrum of SCW-related target genes [24,25,26]. Thus, while the SCW transcriptional network seems to be conserved to some degree, recent evidence suggests that grass-specific nuances exist [27], in line with an expansion of the MYB class of TFs in monocots [28,29]. Recently, MsSND1 and MsSCM1 from *Miscanthus sinensis* have been identified as positive regulators of lignin biosynthesis [30]. However, a detailed analysis of the regulatory properties of MsSND1 and MsSCM1 as well as the impact of these TFs on SCW composition has not been performed yet. In addition, it is unclear whether Miscanthus lower-tier MYB factors are largely redundant or orchestrate a distinct transcriptional response, which would make them interesting targets for tailoring lignin content and/or composition.

Here, we compare two MYB TFs, namely MsSCM1 [30] and the newly identified MsMYB103, with MsSND1, a master switch of SCW formation. In Miscanthus, *MsSCM1* and *MsMYB103* were expressed in tissue undergoing lignification. Ectopically expressed in *N. benthamiana* leaves, MsSND1, MsSCM1, and MsMYB103 were capable of inducing lignin deposition, leading to distinct cell wall compositions. A global expression analysis uncovered the transcriptional responses to *MsSND1*, *MsSCM1*, and *MsMYB103* expression, demonstrating the overlaps and divergences underlying cell wall profiles. Motif enrichment analysis among the differentially expressed genes revealed previously described and novel promoter elements specific for the different transcription factors, potentially facilitating target identification in Miscanthus. Taken together, our results suggest that MsSCM1 and MsMYB103 act as regulators of lignin biosynthesis leading to specific lignin qualities.

## 2. Results and Discussion

### 2.1. Miscanthus MYB103 and SCM1 Are Expressed in Tissues Undergoing Lignification

In order to find potential targets in Miscanthus that allow the engineering of biomass optimized for bioenergy and biomaterials applications, we searched suitable TFs with the ability to regulate SCW formation with a focus on lignin biosynthesis. In Miscanthus, the NAC TF MsSND1 has been previously characterized as a master regulator orchestrating SCW formation, and MsSCM1, a MYB TF related to MYB20/43/85 [18], has been proposed as specific regulator of lignin biosynthesis [30]. We identified a putative MsMYB103 in a Miscanthus transcriptome [7] by a homology-based approach with AtMYB103 as a query sequence. AtMYB103 has been suggested as a key gene regulating the expression of ferulate-5-hydroxylase (F5H), which directs lignin biosynthesis towards formation of syringyl-rich lignin [23] and was initially suggested to exclusively regulate lignin composition. However, MYB103 orthologues of grass species have also been shown to promote cellulose and hemicellulose genes [24,25,26] and AtMYB103 at least associates with cellulose synthase promoter sequences [10,18]. The predicted MsMYB103 protein shares 43.9% identity and 51.9% similarity at the amino acid level with AtMYB103 (Appendix A). An alignment of the R2R3 MYB DNA-binding domain at the N-terminus showed 96% identity at the amino acid level, suggesting similar DNA-binding characteristics between MsMYB103 and AtMYB103.

In monocots, leaf growth is controlled by an intercalary meristem that is situated at the stem node, producing new cells to drive leaf elongation. The continued differentiation along the leaf axis from the immature leaf sheath towards the mature leaf tip results in a linear developmental gradient. Previously, high expression of *MsSND1* was found to coincide with vascular differentiation and SCW formation in Miscanthus leaves [30], suggesting a role of MsSND1 in these processes. To explore the potential involvement of *MsMYB103* and *MsSCM1* in secondary cell wall formation in Miscanthus, the expression of *MsMYB103* and *MsSCM1* was determined together with *MsSND1* along the leaf gradient and visualized in a heat map (Figure 1). The transcript abundance of *MsSND1* was the highest at the leaf base in the first segment, and decreased strongly in the following segments, confirming previous findings [30]. The expression of *MsMYB103* and *MsSCM1* reached its maximum in the second segment and declined markedly in the following segments (Figure 1). The putative orthologues of *MsMYB103* and *MsSCM1* in rice and maize (GRMZM2G325907 and LOC_Os08g05520, as well as GRMZM2G104551 and LOC_Os09g36250, respectively) display a comparable expression pattern [31] in immature leaf, an organ known to show active SCW formation and lignification. This is reminiscent of other putative *MsSND1* target genes [30,32] and supports the roles of *MsMYB103* and *MsSCM1* in SCW formation and lignification.

### 2.2. Ectopic Expression of MsSCM1 and MsMYB103 Lead to Lignin Deposition in N. benthamiana Leaves

In order to investigate the capability of MsSND1, MsSCM1, and MsMYB103 to induce lignification, these TFs were ectopically expressed in *Nicotiana benthamiana* leaves. As previously described [30], MsSND1 induced patterned formation of SCWs in spongy mesophyll, epidermal, and palisade cells reminiscent of xylem elements, which were visualized by Basic Fuchsin staining (Figure 2C, Kapp et al., 2015). In contrast, the ectopic expression of *MsMYB103* resulted in the deposition of lignin in most leaf cells and markedly more intense Basic Fuchsin fluorescence in some cells (Figure 2B), similar to MsSCM1 (Figure 2A, [30]). Notably, lignin deposition lacked the subcellular patterning observed with *MsSND1* expression. This pronounced difference in lignin deposition between *MsSND1-* and *MsMYB103/MsSCM1*-expressing cells confirms distinct transcriptional responses to top-tier NAC TF and lower-tier MYB TF factor expression. It remains elusive as to why some individual cells show stronger lignification after the ectopic expression of MsMYB103/SCM1 than some of their neighboring cells. It is conceivable that only cells with appropriate cell walls can lignify because the incorporation of lignin depends on a suitable polysaccharide matrix [33] and appropriate amounts of lignin precursors such as monolignols [34]. In contrast to the lower-tier MYB TFs, SND1 appears to command the entire differentiation program for xylem cells, including the patterned deposition of the secondary cell wall, which requires the re-organization of the cytoskeleton.

In summary, ectopic expression in *N. benthamiana* leaves revealed that *MsMYB103* and *MsSCM1* are capable of activating the lignin machinery leading to lignin deposition.

### 2.3. Expression of MsSND1, MsMYB103, and MsSCM1 Results in Distinct Secondary Cell Wall Composition

The ability of MsSND1, MsMYB103, and MsSCM1 to regulate cell wall formation prompted us to exploit the *N. benthamiana* system to investigate compositional changes in the cell walls associated with the ectopic expression of the TFs. As expected, in mock-infiltrated *N. benthamiana* leaves, the content of acid-insoluble Klason lignin was low (ca. 15 µg/mg dry weight; Figure 3A). The ectopic expression of *MsSND1*, *AtSND1* [30], *MsSCM1*, and *MsMYB103* elevated Klason lignin content by 2- to 5.5-fold (29 to 82 µg/mg dry tissue). MsSND1 and AtSND1 had a greater effect compared to MsSCM1 and MsMYB103, consistent with the microscopy results obtained with histochemical staining (Figure 2). In contrast, acid-soluble lignin was not significantly affected by the expression of the TFs. The measurement of non-condensed lignin by thioacidolysis demonstrated a higher S/G ratio in *MsMYB103*-expressing tissue, whereas the expression of the other transcription factors resulted in lower ratios, similar to the control (Figure 3B,C). These results are consistent with the described role of Arabidopsis MYB103 as a key regulator of syringyl lignin [23]. However, expressing *PvMYB85A*, a close relative of *MsSCM1*, in switchgrass was capable of elevating the S/G ratio [35], whereas *MsSCM1* did not change the S/G ratio in *N. benthamiana* (Figure 3B,C), highlighting the functional nuances between two closely related genes. Analysis of the structural carbohydrates revealed that the ectopic expression of *MsSND1* and *AtSND1* provokes a strong increase in xylose (Figure 3D), which likely originates from the formation of xylan and/or arabinoxylan in cell walls [30]. The composition of structural carbohydrates in MsSCM1 samples was similar to the control, except for a reduction in glucose, which was observed in all samples. Interestingly, the transient expression of *MsMYB103* led to a small increase in mannose and xylose (Figure 3D), possibly suggesting a regulatory role for *MsMYB103* in hemicellulose formation. Taken together, a chemical analysis of cell walls after the ectopic expression of *MsSND1*, *MsSCM1*, and *MsMYB103* revealed distinct cell wall compositions.

### 2.4. RNA-Seq Analysis Following Expression of MsSND1, MsSCM1, and MsMYB103 Delineates Diverging Transcriptional Landscapes

In order to understand how the expression of *MsSND1*, *MsSCM1*, and *MsMYB103* translates into compositional cell wall changes, we profiled the TF-mediated gene expression changes. The agrobacterium-mediated infiltration of TFs into *N. benthamiana* leaves followed by RNA-seq was shown to be suitable to discover the downstream targets of TFs [36]. Four days post infiltration, we could verify the expression of *MsSND1*, *MsSCM1*, *MsMYB103*, and the putative downstream targets *CCoAOMT* and *XCP1* by q-RT-PCR, suggesting that the experimental approach is suitable for capturing the transcriptional profiles via RNA-seq (Appendix A). An evaluation of the transcriptomes using stringent parameters (adjusted *p*-value < 0.05, at least 4-fold change of expression compared to control) revealed a total of 4716 differentially expressed genes (DEGs) in at least one condition compared to the control (Appendix A). The hierarchically clustered heatmap of normalized counts of DEGs from the control, MsSND1, MsSCM1, and MsMYB103 illustrated very similar expression patterns within biological replicates (Figure 4A). As expected, the transcriptional response to the expression of the master regulator MsSND1 was more extensive (3317 DEGs) than that to MsSCM1 and MsMYB103 expression (2041 and 1392 DEGs, respectively). The samples expressing lower-tier MYB factors shared between 47% and 69% of DEGs with MsSND1 samples (Figure 4B,C), consistent with the described role of MYBs as downstream targets of secondary cell wall NAC master regulators [27].

Gene ontology (GO) term enrichment analyses of DEGs commonly upregulated by MsSND1, MsSCM1, and MsMYB103 (“intersection”) revealed enriched GO terms associated with SCW formation (Figure 5A; Appendix A). The bubble plots in Figure 5 indicate the most enriched GO terms associated with DEGs in the intersection, as well as with total and unique DEG sets of each transcription factor; the complete GO term enrichment analysis can be found in Appendix A. SCWs are formed following the cessation of growth and development, often accompanied by cell death. GO terms linked to photosynthesis, energy metabolism, and nucleotide metabolism were enriched in the intersection of downregulated DEGs (Figure 5B), indicating a tightly controlled switch from active metabolic cells towards specialized cells with a role in mechanical support enacted by both top- and lower-tier TFs in the network. The GO enrichment analysis of DEGs suggests distinct and overlapping functions of *MsSND1*, *MsSCM1*, and *MsMYB103* in lignin biosynthesis and cell wall formation. Interestingly, samples from tissue expressing *MsSCM1* shared only 25% of the upregulated DEGs with tissues expressing MsMYB103, whereas the overlap was even lower (16%) with downregulated DEGs (Figure 4B,C). This suggests that lower-tier MYB factors can have unique targets, possibly contributing to the diversification and fine-tuning of the secondary cell wall transcriptional program.

We then performed de novo motif enrichment analyses in the 2000 bp upstream and 200 bp downstream of the start of the coding sequence of upregulated DEGs using relaxed criteria (adjusted *p*-value < 0.05, at least 2-fold induction, Appendix A). Among genes induced in response to SND1, elements related to the well-known SNBE sequence were significantly enriched. In addition, a SMRE element was enriched in both SND1 and MYB103 DEGs, whereas a MYB consensus element (CAACCA) was overrepresented in SCM1 DEGs. Moreover, previously undescribed motifs are specifically enriched in the regulatory regions of genes induced by the expression of MYB103 (Figure 6). The observation that de novo motif discovery revealed known NAC promoter elements validates this approach for the identification of potentially novel DNA elements.

We focused on the lower-tier MYB factors which are assumed to directly regulate SCW target genes and compared enriched motifs in genes specifically induced by SCM1 (2193 genes), MYB103 (2286 genes), and both (1023 genes), respectively. The identified motifs were largely specific for one of the three classes of DEGs (Figure 7), supporting the notion that these motifs are potentially relevant for the regulation of MYB target genes and that this diversity pf cis-regulatory elements can be harnessed to decipher the distinct transcriptional regulation by different MYB TFs. Thus, the results presented here offer a route for the identification of potential target genes in the Miscanthus genome.

### 2.5. Expression Atlas Highlights Essential Genes of Lignin Biosynthesis

*MsSND1*, *MsSCM1*, and *MsMYB103* expression was found to be associated with distinct cell wall and lignin compositions (Figure 3B). To obtain a detailed perspective on genes involved in lignification that may affect lignin quality, the expression of putative lignin genes from the transcriptomes were plotted in a schematic lignin biosynthesis pathway (Figure 8). The ectopic expression of *MsSND1* and *MsSCM1* elevated the expression of several putative genes of the general phenylpropanoid pathway (*PAL*, *C4H*, and *4CL*), which allocate precursors for monolignol synthesis. It is tempting to speculate that the supply of precursors limits lignin formation, leading to different lignin amounts observed in *MsSND1*, *MsSCM1*, and *MsMYB103* expressing leaves (Figure 3A). Similarly, the high expression of several putative *HCT*, *C3′H*, and *CCoAOMT* genes in MsSND1 and MsSCM1 samples may channel resources for the generation of lignins rich in guaiacyl and syringyl units (Figure 3B and Figure 8). Interestingly, presence of MsMYB103 leads to a less pronounced increase in the expression of these genes, whereas transcripts for F5H and COMT, which redirect monolignol precursors towards sinapyl alcohol, are induced more strongly than by the expression of *MsSCM1*, possibly accounting for the high accumulation of syringyl units in the cell wall lignin (Figure 3B). It has been suggested that AtMYB103 specifically regulates F5H expression to modulate the lignin S:G ratio [23]. However, this direct and specific link could not be established for MYB103 proteins originating from grass species, which were shown to regulate cellulose and hemicellulose [24,25,26]. In line with these results, MsMYB103 seems to orchestrate and finetune SCW formation and lignin biosynthesis more broadly. This notion is supported by the distinct expression profiles of various putative laccases and peroxidases (Figure 8), which may also have implications for lignin composition [37,38,39]. Taken together, the transcriptomic data provide a detailed atlas of the transcriptional network of MsSND1, MsSCM1, and MsMYB103 underlying lignin biosynthesis and SCW formation.

## 3. Materials and Methods

### 3.1. Plasmid Construction

For this study, plasmid constructs were generated via GreenGate cloning [42]. The protein-coding regions of *MsSND1*, *MsSCM1,* and *MsMYB103* were amplified by PCR using appropriate primers with *Bsa*I restrictions site overhang and cDNA from *Miscanthus sinensis* [43]. More information about primer sequences, GreenGate modules, and assembled constructs for ectopic expression can be found in Appendix A.

### 3.2. Gene Expression Analysis

Gene expression analysis was performed according to [30]. Briefly, total RNA was extracted from 30 mg ground tissue from developing leaves approximately 50 cm in size using GeneMatrix Universal RNA Purification Kit (EURx/Roboklon, Berlin, Germany) with on column DNase treatmnent. 1 µg of RNA was reverse transcribed using oligo dT primer either by Superscript III Reverse Transcriptase (Thermo Fisher, Schwerte, Germany) for *N. benthamiana* or by RevertAid First Strand cDNA Synthesis Kit (Thermo Fisher, Schwerte, Germany ) for Miscanthus, according to the manufacturer’s instructions. The quantitative Real-Time PCR was carried out in a total volume of 15 µL containing 2 µL of diluted cDNA, 1 µL 5 µM forward and reverse primer (Appendix A), 0.3 µL of each 10 mM dNTPs, 1:400 diluted SYBR^®^Green I (Sigma-Aldrich, Taufkirchen, Germany), JumpStart™ Taq DNA polymerase in the corresponding buffer, and RNase-free H_2_O. Measurements were observed with a Rotor-Gene Q thermocycler and evaluated by Q Series Software version 2.1.0.9(Qiagen, Hilden, Germany). Gene expression was normalized against UBC and PP2A in Miscanthus and PP2A in *N. benthamiana*.

### 3.3. Nicotiana benthamiana Infiltration

For ectopic expression of transcription factors, the respective GreenGate construct was transformed into *Agrobacterium tumefaciens* ASE (pSOUP^+^). Selected transgenic clones were incubated in liquid LB medium containing the respective antibiotics for two days. The LB medium was replaced with infiltration medium (10 mM MgCl_2_, 10 mM MES, 0.15 mM acetosyringone at pH 5.7). Leaves of 4–6 week-old *N. benthamiana* cultivated in the greenhouse under constant light conditions (150 µmol s^−1^ m^−2^) were co-infiltrated with bacterial suspension of transgenic clones and *A. tumefaciens* C58C1, containing a p14 silencing suppressor [44]. An empty vector control (Appendix A) was co-infiltrated with p14 Agrobacterium as control. Plants were returned to the greenhouse 24 h after infiltration. The infiltration procedure was identical for all experiments, with individual leaves of different plants constituting independent biological replicates.

### 3.4. Tissue Staining and Microscopy

After 5 days of ectopic expression, *N. benthamiana* leaves were embedded in 6% agarose and hand-sectioned with a razor blade. For Basic Fuchsin staining, cross-sections were cleared and fixed for 15 min in methanol, before they were incubated in 10% (*w*/*v*) NaOH at 65 °C for 1 h. Lignin was stained with 0.01% (*w*/*v*) Basic Fuchsin (Sigma-Aldrich, Taufkirchen, Germany) in water for five minutes and washed briefly with 70% (*v*/*v*) ethanol. Cross-sections were mounted in 50% (*v*/*v*) glycerol and imaged on a confocal TCS SP5II microscope (Leica, Mannheim, Germany) equipped with a 40.0 × 1.25 NA objective under 561 nm excitation and 593/40 nm emission. ImageJ v1.52 software (https://imagej.nih.gov/ij/, accessed on 12 November 2021) was used to produce orthogonal sections and scale bars.

### 3.5. Cell Wall Analysis

*N. benthamiana* leaves were harvested 5 days post infiltration. Plant samples were dried for 2 days at 40 °C after removing non-infiltrated tissue such as midribs and mature veins. Dry plant material was ground with a Wiley mill and subsequently Soxhlet extracted with hot acetone (70 °C) for 24 h. Extractive-free plant material was used to determine Klason lignin content and structural carbohydrate concentration according to [45]. The monolignol composition was determined by thioacidolysis as described in [46] with minor modifications. Since the overall lignin content was relatively low, lignin monomers were extracted from 20 mg of extractive-free plant tissue and 2 µL was injected into the gas chromatograph.

### 3.6. Transcriptome Analysis

To minimize the effects of natural transcriptional differences between *N. benthamiana* plants, three single developing leaves of three different plants were infiltrated with all three TFs and the control according to the scheme illustrated in Appendix A, constituting three independent biological replicates. In order to capture the high gene expression of transcriptional networks, leaf tissue was harvested 4 days post infiltration and immediately frozen and ground in liquid nitrogen. Total RNA was extracted from 50 mg ground leaf tissue using a GeneMatrix Universal RNA Purification Kit (EURx/Roboklon; Berlin, Germany) with on-column DNase treatment. Library preparation was performed by GATC Biotech (Konstanz, Germany) using proprietary methods. Subsequently, the strand-specific cDNA library was sequenced by GATC Biotech on Illumina HiSeq 4000 instruments in 150 bp paired-end mode.

The re-annotated *N. benthamiana* transcriptome [40] was downloaded from ORA (https://ora.ox.ac.uk/objects/uuid:f09e1d98-f0f1-4560-aed4-a5147bc7739d, accessed on 12 November 2021). The index of the reference transcriptome was built using Bowtie2 [47] and paired-end RNA sequencing reads were aligned to the reference transcriptome using RSEM [48] on Galaxy (RSEM-Bowtie2 v0.9.0). Differentially expressed genes (DEGs) between control- and TF-infiltration lines were identified with edgeR using read counts for each transcript and TMM normalization. Genes with an adjusted *p*-value < 0.05 and a minimum four-fold change were considered to be differentially expressed (Appendix A). The heatmap was visualized with normalized counts (logCPM) that were hierarchically clustered by complete linkage and Euclidean distance using heatmap.2 in gplots. GO enrichment was performed based on the hypergeometric test and Benjamini–Hochberg correction. Enrichment was calculated on go.obo file from Geneontology website (http://geneontology.org/docs/download-ontology, accessed on 12 October 2021), and the *N. benthamiana* gene annotation based on Supplemental_dataset_5_NbC_Annotation_all from ref 40, downloaded from ORA (https://ora.ox.ac.uk/objects/uuid:f09e1d98-f0f1-4560-aed4-a5147bc7739d, accessed on 12 November 2021). The bubble map is based on the 10 most significantly enriched biological process categories of the intersection of co-regulated DEGs, total DEGs, and DEGS unique for each transcription factor.

### 3.7. Accession Numbers

AmMYB330 (P81393), AtMYB20 (AT1G66230.1), AtMYB43 (AT5G16600.1), AtMYB46 (AT5G12870.1), AtMYB58 (AT1G16490.1), AtMYB61 (AT1G09540.1), AtMYB63 (AT1G79180.1), AtMYB83 (AT3G08500.1), AtMYB85 (AT4G22680.1), AtMYB103 (AT1G63910.1), BdSWAM1 (KQK09372), EgMYB1 (CAE09058), EgMYB2 (CAE09057), GhMYBL1 (KF430216), MsSCM1 (KY930621), MsSCM2 (MF996502), MsSCM3 (KY930622), MsSCM4 (MF996501), MsMYB103 (MK704407), OsMYB103L (LOC_Os08g05520.1), PbrMYB169 (MG594365), PtMYB1 (AAQ62541), PtMYB4 (AAQ62540), PtMYB8 (ABD60280), PtoMYB92 (KP710214), PtrMYB2 (AGT02397), PtrMYB3 (AGT02395), PtrMYB10 (XP_002298014), PtrMYB20 (AGT02396), PtrMYB21 (AGT02398), PtrMYB128 (XP_002304517), PtrMYB152 (POPTR_0017s02850), PvMYB63A (Pavir.Bb02654), PvMYB85A (Pavir.Gb00587), and SbMYB60 (Sobic.004G273800).

## 4. Conclusions

In order to explore the potential of lignocellulosic biomass as a sustainable resource, lignin engineering has become a focus of research with the goal to either decrease recalcitrance or to produce more valuable lignin with enhanced utility. The genus Miscanthus has gained momentum as a prospective lignocellulosic feedstock, but the mechanisms underlying lignin and SCW formation have only been poorly investigated. In this study, we identify and characterize three TFs, namely MsSND1, MsSCM1, and MsMYB103, as regulators involved in lignin biosynthesis associated with distinct cell wall compositions that present interesting targets for specific cell wall manipulations. In order to bypass the challenging and time-consuming transformation procedures of Miscanthus, infiltration-RNA-seq was investigated as a suitable method to uncover the transcriptional regulation of TFs involved in SCW biosynthesis and beyond. The extensive transcriptomic data in response to the expression of *MsSND1*, *MsSCM1*, and *MsMYB103* provided in this study offer the opportunity to gain a deeper understanding of lignin formation. De novo motif discovery in the regulatory regions of genes induced by the three transcription factors yielded previously described and novel promoter elements, thus confirming the validity of the approach and providing entry points for identifying potential genes of interest in Miscanthus. Despite intensive research focusing on the lignin biosynthetic pathway, it is still a matter of debate how monolignols are trafficked to the cell wall and which are the driving forces behind this process. It also remains elusive if glycosylated monolignols play an essential role as storage and/or transport molecules. The comparison of the three transcriptomes allowed for the identification of potential candidate genes involved in storage (UGTs, β-Glucosidases), transport (ABC transporter) and/or oxidative polymerization (laccases, peroxidases), providing a valuable starting point for future research.

## Figures and Tables

**Figure 1 ijms-22-12395-f001:**
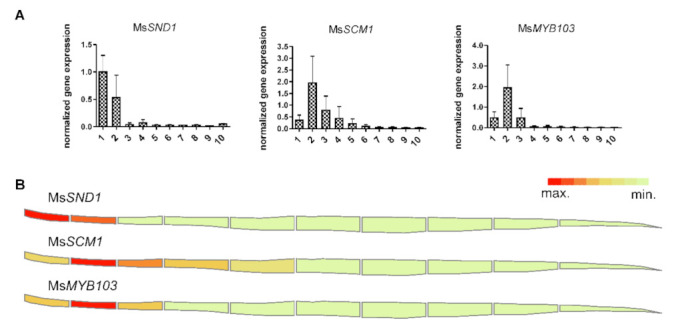
Expression of MsSND1, MsSCM1, and MsMYB103 is associated with tissues undergoing lignification. Developing leaves of ~50 cm length were dissected into ten sections spanning from leaf base at the left side to leaf tip on the right side. Relative gene expression values are calculated from three biological replicates (**A**), which are normalized against two reference genes (UBC and PP2A) and visualized as a heat map (**B**).

**Figure 2 ijms-22-12395-f002:**
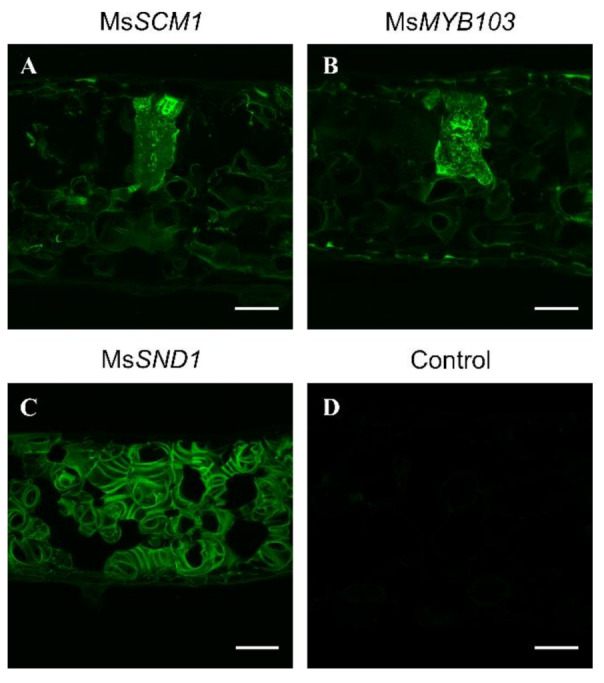
Lignin formation induced by ectopic expression of MsSCM1 (**A**), MsMYB103 (**B**), and MsSND1 (**C**) in *N. benthamiana* leaves. After 5 days of ectopic expression, leaf cross-sections were stained with Basic Fuchsin to observe fluorescence under a confocal microscope. Infiltration of the empty vector served as control (**D**). Scale bar—50 µm.

**Figure 3 ijms-22-12395-f003:**
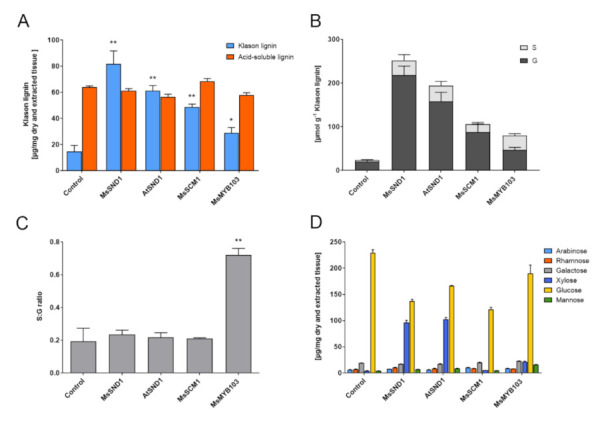
Change in cell wall composition induced by ectopic expression of MsSND1, MsSCM1, and MsMYB103 in *N. benthamiana* leaves. (**A**) Lignin content was determined by Klason analysis (acid-insoluble lignin) and by OD_205_ (acid-soluble lignin); (**B**,**C**) ratio of syringyl and guaiacyl monomer release from thioacidolysis products detected by gas-chromatography; (**C**,**D**) structural carbohydrate compositions of insoluble extracts were determined by high-pressure liquid chromatography (HPLC), as µg/mg of dry cell walls. As a control, an empty vector construct was infiltrated in *N. benthamiana* leaves. The bars represent mean values from two (**B**,**C**) or three (**A**,**D**) biological replicates + SE including two technical replicates for (**A**,**D**). Asterisks in (**A**,**C**) indicate statistically significant difference with *p* < 0.05 (*) or *p* < 0.01 (**) according to Mann-Whitney U test.

**Figure 4 ijms-22-12395-f004:**
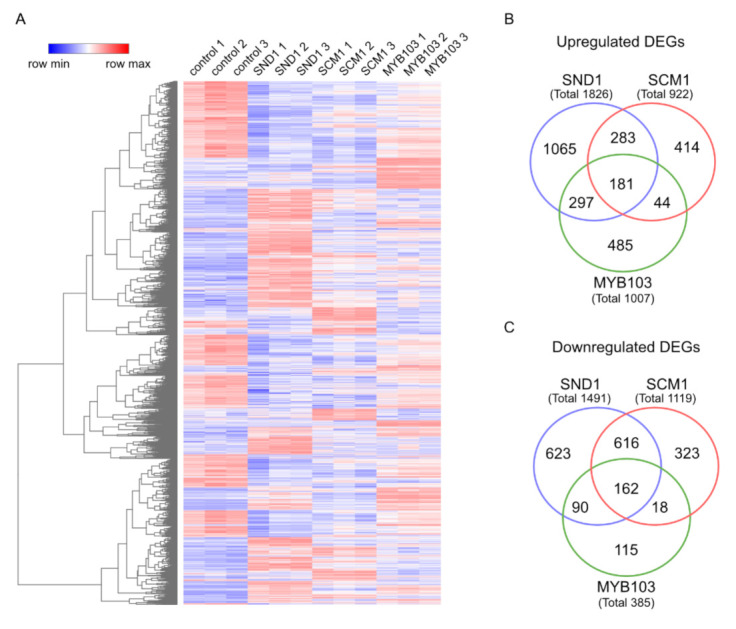
Transcript profiling of transcriptional network regulated by MsSND1, MsSCM1, and MsMYB103. (**A**) Heatmap of normalized counts (logCPM) of the 4716 genes that were differentially expressed (4-fold cutoff) in at least one sample compared to control infiltrations with an empty vector. Rows were hierarchically clustered by complete linkage and Euclidean distance. (**B**,**C**) Venn diagram of upregulated (**B**) and downregulated (**C**) DEGs with adjusted *p*-value < 0.05 and a minimum 4-fold change.

**Figure 5 ijms-22-12395-f005:**
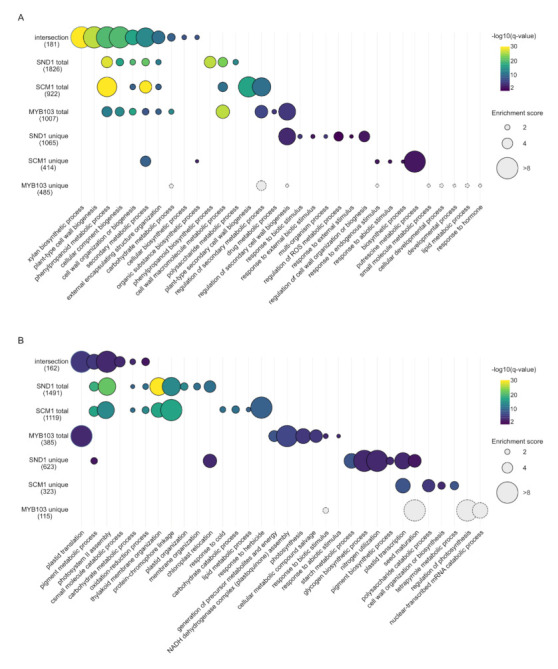
Bubble plots indicating shared and distinct GO terms enriched in different sets of DEGs. (**A**) Upregulated DEGs. (**B**) Downregulatd DEGs. Colors represent the −log_10_ transformation of the adjusted *p*-value (*q*-value), the size of the circles represents enrichment score of the corresponding GO term. Enrichment score is calculated by dividing the percentage of genes associated with a given GO term in the respective DEG list by the percentage of associated genes in the genome. Dashed circles indicate non-significant enrichment based on adjusted *p*-values. The ten most significantly enriched GO terms in each DEG list were used to construct the plot. Categories represented by fewer than three genes were excluded. See Appendix A for complete datasets.

**Figure 6 ijms-22-12395-f006:**
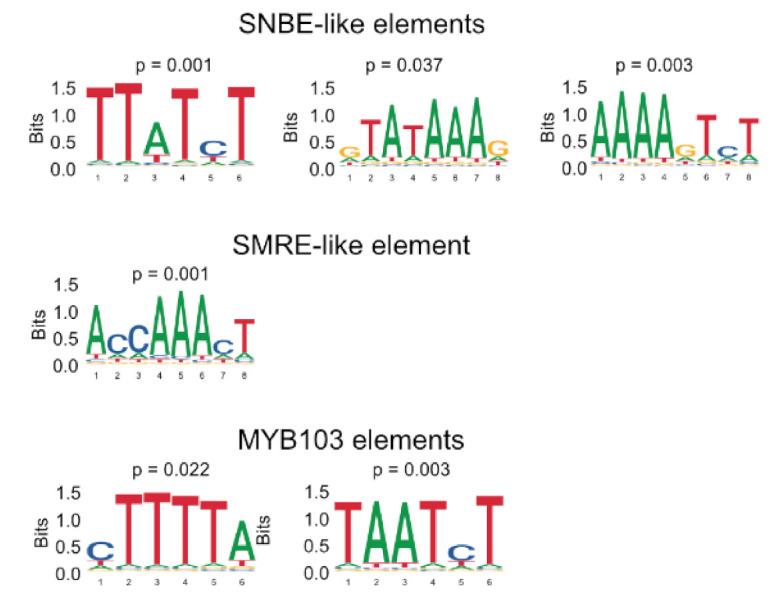
Motifs discovered and overrepresented in genes induced by expression of SND1 and MYB103.

**Figure 7 ijms-22-12395-f007:**
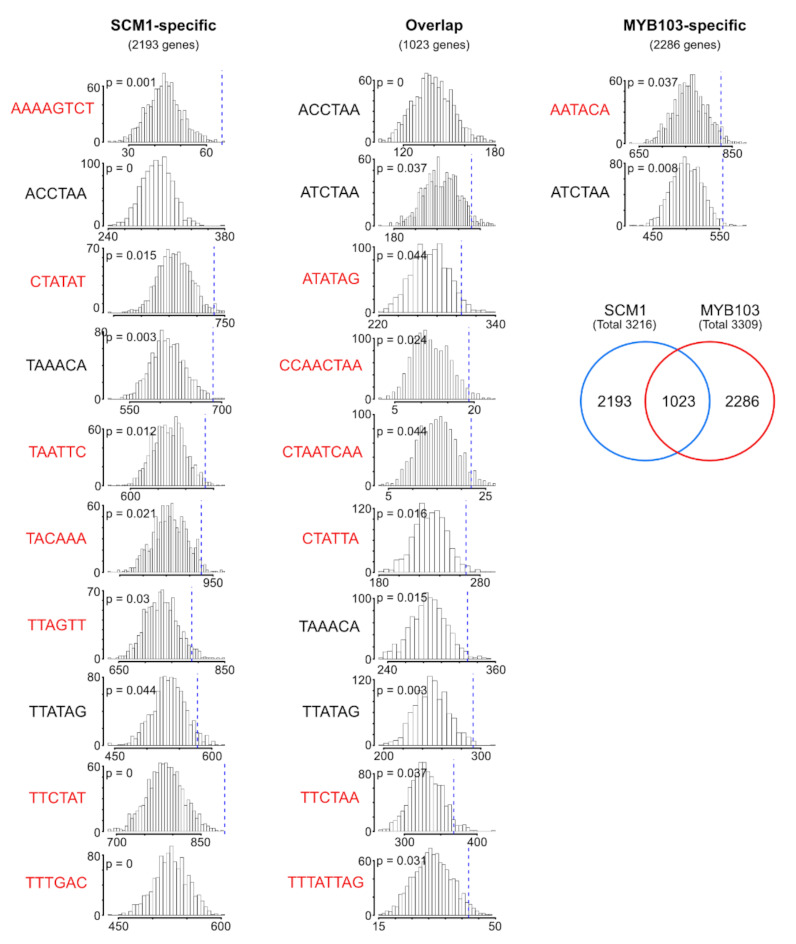
Comparison of motifs discovered in the genes induced by expression of the lower-tier MYB factors SCM1 and MYB103. Motifs were discovered in regions 2000 bp upstream and 200 bp downstream of the start codon of genes significantly and more than 2-fold induced by expression of either MYB TF, or by both, followed by enrichment analyses. Histograms depict occurrence of the indicated motifs in DEG regulatory regions or randomly sampled regions in 1000 iterations. Comparison of motifs derived from the MYB factors demonstrates that the majority of motifs (red) are specific to one of the TFs, in line with the limited overlap of DEGs.

**Figure 8 ijms-22-12395-f008:**
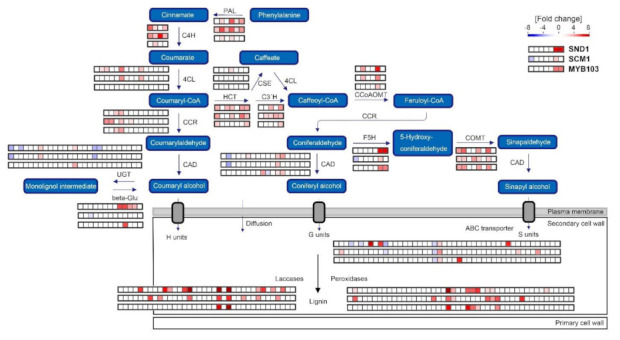
A schematic overview of lignin biosynthesis pathway and DEGs regulated by MsSND1, MsSCM1, and MsMYB103. The fold change of putative genes involved in lignin biosynthesis according to functional annotation by [40]) are represented in colored squares, generated by MapMan software [41]. The rows of squares visualize the three samples, MsSND1, MsSCM1, and MsMYB103, respectively. Red indicates upregulation, while blue indicates downregulation. The key in the upper right part of Figure uses the F5H expression data as example. 4CL, 4-coumarate CoA ligase; beta-Glu, β-Glucosidase; C3′H, p-coumaroyl shikimate 3′-hydroxylase; C4H, cinnamate 4-hydroxylase; CAD, cinnamyl alcohol dehydrogenase; CCoAOMT, caffeoyl CoA O-methyltransferase; CCR, cinnamoyl CoA reductase; COMT, caffeic acid O-methyltransferase; CSE, caffeoyl shikimate esterase; F5H, ferulate 5-hydroxylase; HCT, hydroxycinnamoyl CoA:shikimate hydroxycinnamoyl transferase; PAL, phenylalanine ammonia lyase; and UGT, UDP-Glucuronosyltransferase.

## Data Availability

The RNA sequencing datasets obtained in this study are available in the Sequence Read Archive (SRA) database under BioProject PRJNA529062. https://www.ncbi.nlm.nih.gov/bioproject/PRJNA529062 ( accessed on 12 November 2021).

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
