# Peer review of "Distinct and Overlapping Functions of Miscanthus sinensis MYB Transcription Factors SCM1 and MYB103 in Lignin Biosynthesis"

_ijms, 2021, doi:10.3390/ijms222212395_

Round 1
Reviewer 1 Report
- Authors have done the entire analysis on the infiltrated tobacco. Although they have mentioned the Miscanthus transformation as a reason but looking at the literature, Miscanthus transformation is fairly straightforward and reasonably fast too.
- It would be interesting to see what kind of Go terms are enriched in the uniquely up-regulated DEGs for each TFs in fig 4b?
- In methods, the authors have not mentioned any information about the leaf dissection for the expression studies and what was the size/criteria to dissect the leaves. How they were processed, etc.
- AtSND1 cloning or transformation is not mentioned anywhere in the method section but all of a sudden its appearing in results.
- Authors should carefully correct the manuscript for the fold change criteria mentioned at different places for the same analysis. Line 233 suggests a 4-fold cutoff whereas figure 4 for the same result says 2 fold. Also, add the one-way clustering dendrogram to the heatmap. In the absence of any clear pattern, the heatmap does not look like has been clustered for the genes.
- Sum total of SCM1 DEGs from Fig 4B and 4D is not the same as whats mentioned in line 239 (2241)
- Line 405: what was the 4th TF? Transcriptome results only show data related to 3 TFs.
- Fig6, authors need to better explain how they got these numbers, and are they still only DEGs? Once they correct the manuscript for comment 4, this might be clear.
- Authors should mention in methods whether or not the total RNA was treated with DNase?
- Please mention details of figure sub-panels in the figure legend section for Figures 1 and 2.
Author Response
Reviewer 1
- Authors have done the entire analysis on the infiltrated tobacco. Although they have mentioned the Miscanthus transformation as a reason but looking at the literature, Miscanthus transformation is fairly straightforward and reasonably fast too.
>> Transformation of Miscanthus is unfortunately neither fast nor straightforward. We are aware of this limitation of our study but remain confident that it remains useful for the future.
- It would be interesting to see what kind of Go terms are enriched in the uniquely up-regulated DEGs for each TFs in fig 4b?
>>We thank the reviewer for this suggestion. We have re-analysed the DEG list for GO term enrichment and represent the shared and distinct terms using bubble plots, included as new figure 5.
- In methods, the authors have not mentioned any information about the leaf dissection for the expression studies and what was the size/criteria to dissect the leaves. How they were processed, etc.
>>The approach follows our earlier publication (Golfier et al., 2017) and specifics of leaf dissection are given there (Fig2). However, we have amended the methods section to include information on leaf size; The sections indicated in Fig1 B are roughly proportional to the actual segments.
- AtSND1 cloning or transformation is not mentioned anywhere in the method section but all of a sudden its appearing in results.
>>This construct is described in Golfier et al., 2017. We apologize for omitting the reference. This is now fixed in the revised manuscript
- Authors should carefully correct the manuscript for the fold change criteria mentioned at different places for the same analysis. Line 233 suggests a 4-fold cutoff whereas figure 4 for the same result says 2 fold. Also, add the one-way clustering dendrogram to the heatmap. In the absence of any clear pattern, the heatmap does not look like has been clustered for the genes.
>>We thank the reviewer for the observation and suggestions. The heat map and GO enrichment analysis are indeed based on a 4-fold cut off, whereas motif discovery in regulatory regions uses DEGs defined with a two-fold cut off. The mislabelling in the legend for Figure 4 is corrected in the revised manuscript. In addition, the reviewer is of course right that the heatmap presented was not actually hierarchically clustered. The revised version of the manuscript now contains a clustered heatmap as Figure 4A
- Sum total of SCM1 DEGs from Fig 4B and 4D is not the same as whats mentioned in line 239 (2241)
>> Line 233 contained a typo and should read 2014, which is corrected in the revised manuscript.
- Line 405: what was the 4th TF? Transcriptome results only show data related to 3 TFs.
>>The reviewer is right and the revised manuscript refers to three TFs and control.
- Fig6, authors need to better explain how they got these numbers, and are they still only DEGs? Once they correct the manuscript for comment 4, this might be clear.
>>We hope this indeed now clear due to correction of Figure 4, which is based on 4-fold cut off.
- Authors should mention in methods whether or not the total RNA was treated with DNase?
>>RNA was indeed treated with DNase, which is now mentioned in the methods section
- Please mention details of figure sub-panels in the figure legend section for Figures 1 and 2.
>>The manuscript was revised according to the reviewer’s suggestion.
Reviewer 2 Report
This manuscript is interesting. The information provided is novel, the authors have conducted a careful research plan and their conclusions are well supported by the results. I see no major flaws. I only thing that the introduction and the discussion parts could be reduced a bit.
I therefore recommed a minor revision.
Author Response
Reviewer 2.
I only thing that the introduction and the discussion parts could be reduced a bit.
>>We thank the reviewer for the positive assessment of our work. Regarding the introduction, we felt that shortening would come at the expense of accessibility and therefore refrained from shortening. There is no formal discussion section in the manuscript, but we shortened the section on GO enrichment slightly.
Reviewer 3 Report
This paper addressed exciting and essential questions about lignin biosynthesis and secondary cell wall. Authors test two genes from Miscanthus (Silvergrass) - MsSCM1 and MsMYB103 after their expression in Nicotiana and shown that they act as regulators of lignin biosynthesis leading to changes in lignin qualities.
Line 61: „vital to understand lignin biosynthesis“ – maybe better to add „mechanism of biosyntehsis“
Line 70: „breeding has focused on the development of molecular markers“? Breeding can not focused on developmnet of molecular marker“, they may require such marker! please, re-formulate.
Line 141: intecalary merisrem also produced new cells, not only „driving leaf elongation2.
Figure 1: it will be nice to show segment size and somehow marker of cell division (optionaly).
Line 164: this is not ectopic expression because not all cell expressed these gene. Infoltartion can never reach 100% and it will be nice in fiture the estimate at least approximately how much cell expressed gene.
Please, also provide infiltration protocol and post-infiltartion cultivation. For the lignin you need to have sufficinet illumination to get procursor as photosyntehsis results or used medium with precursors like sucrore, glucose etc. But there are no details written here.
Line 268> de novo must be in italics.
Author Response
Reviewer 3.
This paper addressed exciting and essential questions about lignin biosynthesis and secondary cell wall. Authors test two genes from Miscanthus (Silvergrass) - MsSCM1 and MsMYB103 after their expression in Nicotiana and shown that they act as regulators of lignin biosynthesis leading to changes in lignin qualities.
Line 61: „vital to understand lignin biosynthesis“ – maybe better to add „mechanism of biosyntehsis“
>>We thank the reviewer for the suggestion and have revised the manuscript accordingly.
Line 70: „breeding has focused on the development of molecular markers“? Breeding can not focused on developmnet of molecular marker“, they may require such marker! please, re-formulate.
>> We have rephrased this statement in the revised version of the manuscript to “Moreover, Miscanthus breeding would benefit from molecular markers to facilitate selection of varieties with improved biomass yield and/or composition”
Line 141: intecalary merisrem also produced new cells, not only „driving leaf elongation2.
>>This sentence has been amended and now also mentions production of new cells.
Figure 1: it will be nice to show segment size and somehow marker of cell division (optionaly).
>>Leaf size is indicated in Golfier et al., and now also mentioned in the method sections. Segment sizes are proportional to what is indicated in Figure 1B.
Line 164: this is not ectopic expression because not all cell expressed these gene. Infoltartion can never reach 100% and it will be nice in fiture the estimate at least approximately how much cell expressed gene.
>>We use ectopic in the sense of “out of place” indicating that expression occurs in cells that would not normally form SCWs.
While we agree that expression can not be assumed to be induced in all cells, most mesophyll and epidermis cells show evidence of expression (fluorescence signal compared to control infiltration).
Please, also provide infiltration protocol and post-infiltartion cultivation. For the lignin you need to have sufficinet illumination to get procursor as photosyntehsis results or used medium with precursors like sucrore, glucose etc. But there are no details written here.
>>This information is now contained in the methods sections. Plants were returned to continuous light conditions 24 hours after infiltration.
Line 268> de novo must be in italics.
>>This is now corrected in the revised manuscript.
Round 2
Reviewer 1 Report
I thanks authors for acknowledging the comments and successfully answered them. I happy to endorse the manuscript for publication.